# A Study of Drag Reduction on Cylinders with Different V-Groove Depths on the Surface

**Jiyang Qi** [1], **Yue Qi** [1], **Qunyan Chen** [2] **and Fei Yan** [1,*]

1   School of Mechanical Engineering, Jiangsu University of Science and Technology, Zhenjiang 212000, China; jyqi@just.edu.cn (J.Q.); qiyue111@stu.just.edu.cn (Y.Q.)
2   Department of Metallurgy and Automotive Engineering, Shandong Vocational College of Industry, Zibo 256414, China; chenqy1214@just.edu.cn
*   Correspondence: yanfei@just.edu.cn; Tel.: +86-13862456975

**Abstract:** In this study, the drag reduction effect is studied for a cylinder with different V-groove depths on its surface using a $k\text{-}\omega/SST$ (Shear Stress Transport) turbulence model of computational fluid dynamics (CFD), while a particle image velocimetry (PIV) system is employed to analyze the wake characteristics for a smooth cylinder and a cylinder with different V-groove depths on its surface at different Reynolds numbers. The study focuses on the characteristics of the different V-groove depths on lift coefficient, drag coefficient, the velocity distribution of flow field, pressure coefficient, vortex shedding, and vortex structure. In comparison with a smooth cylinder, the lift coefficient and drag coefficient can be reduced for a cylinder with different V-groove depths on its surface, and the maximum reduction rates of lift coefficient and drag coefficient are about 34.4% and 16%, respectively. Otherwise, the vortex structure presents a complete symmetry for the smooth cylinder, however, the symmetry of the vortex structure becomes insignificant for the V-shaped groove structure with different depths. This is also an important reason for the drag reduction effect of a cylinder with a V-groove surface.

**Keywords:** flow around a cylinder; V-groove; drag reduction; particle image velocimetry; vortex structure

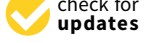



## 1. Introduction

Flow around a cylinder has always been a classical problem in the field of fluid mechanics, one which contains complex flow phenomena such as flow separation, vortex shedding, and wake evolution. In addition, vortex-induced vibration is easily generated when fluid flows through a cylinder, such as marine leggings and marine risers. Therefore, it is of great significance to study the characteristics of the flow field in the wake of the flow around a cylinder.

Many scholars are interested in the mechanism of flow around a cylinder and related flow field characteristics. These phenomena can be used to explain the mechanism of vibration reduction and drag reduction. In general, drag reduction methods can be classified into active control and passive control. Both of them have been proposed to alter or weaken the pattern of vortex shedding and achieve the purpose of drag reduction and vibration suppression. Martin [1] used physical model tests to explore the relationship of vibrations between the downstream and transverse flow of a cylinder. The results demonstrate that the transverse vibration frequency is half of the downstream vibration frequency within a certain speed range. Chen [2] conducted a two-dimensional numerical simulation of the flow around a cylinder and a circular arc cylinder at different angles. The results show that the galloping force coefficient of the circular arc cylinder has an area less than zero, and galloping may occur. Mario et al. [3] used a two-dimensional numerical simulation method to study the perturbation of a single slit on the baseball flow at different angles of attack. The results show that joints located at the angles of 30°, 85° and 90° have a greater impact

on the aerodynamic coefficient and wake. Xu and Chen [4] used the LES (Large Eddy Simulation) method to numerically simulate the characteristics of the flow field around the rotating cylinder and the change of the free shear layer. The results show that the change of the position of the vortex below the rear side of the rotating cylinder has an important effect on the lift and on changes of the free shear layer in the wake area. Yan and Yang [5] performed numerical simulations and experiments on smooth cylinders and cylinders with pits. The results show that the concave structure can effectively reduce the resistance of the cylinder, and the maximum drag reduction rate can reach 19%. Farhana and Muhammad studied the laminar cross flow on smooth and longitudinally slotted cylindrical surfaces by means of numerical simulation. It is found that the grooves decreased viscous drag significantly (up to 30%) compared with pressure drag and the grooves reduced the extent of the recirculation zone [6]. Baek and Karniadakis [7] suppressed the occurrence of VIV (Vortex-induced vibration) by opening a gap inside the cylinder, allowing fluid to pass through the gap, then disrupting the periodic shedding of the wake vortex by the impact force of the jet, wherein an optimum gap size when $Re$ = 500 was determined. In the aspect of numerical simulation, Kakimpa et al. [8,9] employed a computational fluid-dynamics, rigid-body dynamic coupling model to simulate the bidirectional fluid—solid coupling of the eccentric elliptical cylinder around the flow oscillation, which provides a basis for self-excited rotation to obtain low-flow current energy. Zhu and Zhang [10] adopted the slicing method to investigate the effect of changing the cross-section of a riser and placing the moving wave wall on the flow surface of the riser back to suppress the VIV of the flexible riser. Studies have shown that setting a moving wave wall at a certain frequency reduces the transverse amplitude of the riser to less than 0.05 times the diameter of the riser. El-Makdah and Oweis [11] were inspired by the shape of cactus to create an experimental model with a cactus shape and conduct a wind tunnel test. It was found that the closer the wake vortex is to the wall, the smaller the cutting speed is. Oruc et al. [12] used PIV to demonstrate the vortex structure of a single-cylinder wake region with different Reynolds numbers that arose due to the shear layer decoupling from the cylindrical surface. It was proved by investigating the vorticity, Reynolds stress and turbulent energy of the flow field that arranging a water droplet-shaped mesh structure around the cylinder suppresses the formation of a vortex during the flow around the cylinder. Xu et al. [13] performed VIV suppression research on the cylindrical surfaces with square columns or circular columns and found that the columns of square and circular structures were greatly reduced in a strain, displacement amplitude and mean frequency. Yu et al. [14] analyzed the influence of different chamfering radii on the flow around the square column. It was found that drag coefficient and lift coefficient with chamfer on the surface were significantly reduced, by 50% and 53% respectively, effectively controlling the flow characteristics around the square column.

The V-groove structure originated from the study of shark skin in bionics. Previous studies have shown that the V-groove structure would generate a flow vortex at the bottom of the groove under the action of a certain flow field, as shown in Figure 1. The vortex at the bottom of the V-groove is analogous to how "rolling bearings" can reduce the friction resistance between a fluid and the surface of a structure to alter the flow characteristics near the wall surface [15]. Compared with adding an auxiliary structure on the cylinder surface, the V-groove structure has the following advantages: the V-groove structure does not change the shape of the original structure to a great extent, and the V-groove structure can be used in conjunction with other drag reduction methods; it is therefore omnidirectional to the flow control.

In this study, the drag reduction effect and wake characteristics are studied for a cylinder with different V-groove depths on its surface using a k-ω/SST turbulence model of computational fluid dynamics (ANSYS 15.0, Customer # 503068) and particle image velocimetry (PIV) technology at the different Reynolds numbers. The study focuses on the characteristics of the different V-groove depths on lift coefficient, drag coefficient, the velocity distribution of flow field, pressure coefficient, vortex shedding, and vortex structure.

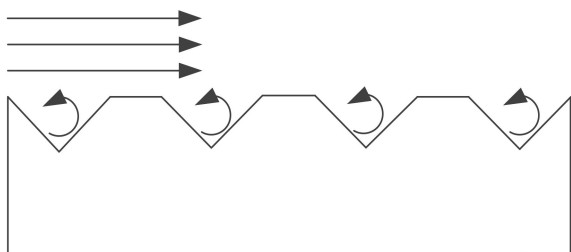

**Figure 1.** Schematic of vortex at the bottom with the groove.

## 2. Mathematical Model Parameter Settings and Experimental Equipment

### 2.1. Control Equation

Navier–Stokes equations (N-S equations) are used to solve a turbulence problem which consumes huge computational resources. A practical way to avoid huge computational demand on turbulence modeling is the implementation of RANS equations (Reynolds-averaged N-S equations). These are different from direct numerical simulation (DNS), as DNS does not directly solve the instantaneous N-S equations but reflects these instantaneous pulsations in the time-homogenized equations using a turbulence model. The Reynolds average equation can be expressed by the following formula

$$\frac{\partial \overline{u_i}}{\partial x_i} = 0 \tag{1}$$

$$\frac{\partial \rho \overline{u_i}}{\partial t} + \frac{\partial \rho \overline{u_i u_j}}{\partial x_j} = -\frac{\partial \overline{p}}{\partial x_i} + \mu \, \partial^2 / \partial x^2{}_i \overline{u_i} - \frac{\partial \rho \overline{u'_i u'_j}}{\partial x_j} \tag{2}$$

$$u'_i = u_i - \overline{u_i}; u'_j = u_j - \overline{u_j} \tag{3}$$

$$-\rho \overline{u'_i u'_j} = \mu_t \left( \frac{\partial u_i}{\partial x_j} + \frac{\partial u_j}{\partial x_i} \right) - \frac{2}{3} \rho k_t \delta_{ij} \tag{4}$$

where $\overline{u_i}$ and $\overline{u_j}$ are time-averaged velocities, $u'_i$ and $u'_j$ represent fluctuating velocities, $u$ is fluid velocity (m/s), $i, j$ are the subscripts, which can be 1, 2 and 3, representing the direction of the coordinate axis, $\mu_t$ is the turbulent viscosity coefficient, $k$ is the turbulent kinetic energy (TKE, Turbulent Kinetic Energy), a measure of the intensity of turbulence) which is directly related to the momentum and energy transfer in the boundary layer. Further, $\delta_{ij}$ is the Kronecker delta symbol, and $-\rho \overline{u'_i u'_j}$ is the Reynolds stress term, which leads to the equations being unclosed, so a turbulence model is needed to close the stress term.

### 2.2. Turbulence Model

The turbulence model in this paper is a shear stress transport (SST) $k$-$\omega$ two-equation model, which is an improved model based on the $k$-$\omega$ model. The $k$-$\omega$ model is adopted at the near-wall. Meanwhile, a high Reynolds number $k$-$\omega$ model is used at the far wall of fully developed turbulence. The transport equation is:

$$\frac{\partial(\rho k)}{\partial t} + \frac{\partial(\rho k u_i)}{\partial x_i} = \frac{\partial}{\partial x_j} \left[ \left( \mu + \frac{\mu_t}{\sigma_k} \right) \frac{\partial k}{\partial x_j} \right] + \widetilde{P}_k - \beta^* \rho \omega k \tag{5}$$

$$\frac{\partial(\rho \omega)}{\partial t} + \frac{\partial(\rho \omega u_i)}{\partial x_i} = \frac{\partial}{\partial x_j} \left[ \left( \mu + \frac{\mu_t}{\sigma_\omega} \right) \frac{\partial \omega}{\partial x_j} \right] + P_\omega - \beta \rho \omega^2 + 2\rho \frac{(1 - F_1)}{\omega \sigma_{\omega,2}} \frac{\partial k}{\partial x_j} \frac{\partial \omega}{\partial x_j} \tag{6}$$

Here, $\sigma_k$ (=1.176) and $\sigma_\omega$ (=1.168) are the turbulent Prandtl numbers about $k$ and $\omega$, where $\omega$ is defined as the ratio $\varepsilon / k_t$, fluid density $\rho$ = 1025 kg/m$^3$, fluid viscosity $\mu$ = 0.001025 Pa·s; $\widetilde{P}_k$ is the effective rate of turbulent kinetic energy generation, which varies with the fluid velocity at different times and different positions; $P_\omega$ is the rate

of turbulent dissipation, which varies with the velocity at different times and different positions; $\beta^*(\beta^* = 0.09)$ is the model constant; $F_1$ is the mixing function, which represents the parameters in the SST $k$-$\omega$ turbulence model [16].

Compared with other standard models, the $k$-$\omega$-$SST$ model is sensitive to flow separation and pressure gradient changes and can well capture the wake characteristics [17] near the wall.

### 2.3. Geometric Models and Boundary Conditions

To examine the influence of different V-groove depths on the flow characteristics of the cylinder, $Re = 3 \times 10^4$ was selected for numerical simulation. $Re = 3 \times 10^4$ is chosen because the water environment where the pile legs of offshore platforms are usually in the sub-critical Reynolds number or higher Reynolds number working conditions. The model of the fluid area and the coordinate system is shown in Figure 2a. The diameter of the cylinder is $D = 0.02$ m. The length of downstream (the x-direction) is 20$D$, the width of the vertical flow direction (the y-direction) is 10$D$, and the height of the extension direction (the z-direction) is $\pi D$. The center of the cylinder is 5$D$ away from the inlet and 5$D$ away from the two walls. Figure 2b shows the schematic diagram of the V-groove structure, where h is the groove depth, $\theta$ is the groove angle ($\theta$ is a constant, $\theta^\circ = 60$), and the cylinder is uniformly arranged with 16 columns of V-shaped grooves of the same size. For the convenience of description and analysis, the calculation examples of different groove parameters are now defined: *case0* represents the smooth cylinder. *case1*, *case2*, *case3*, and *case4* respectively represent cylindrical surfaces with V-groove depths of $h/D = 0.02$, $h/D = 0.03$, $h/D = 0.04$, and $h/D = 0.05$.

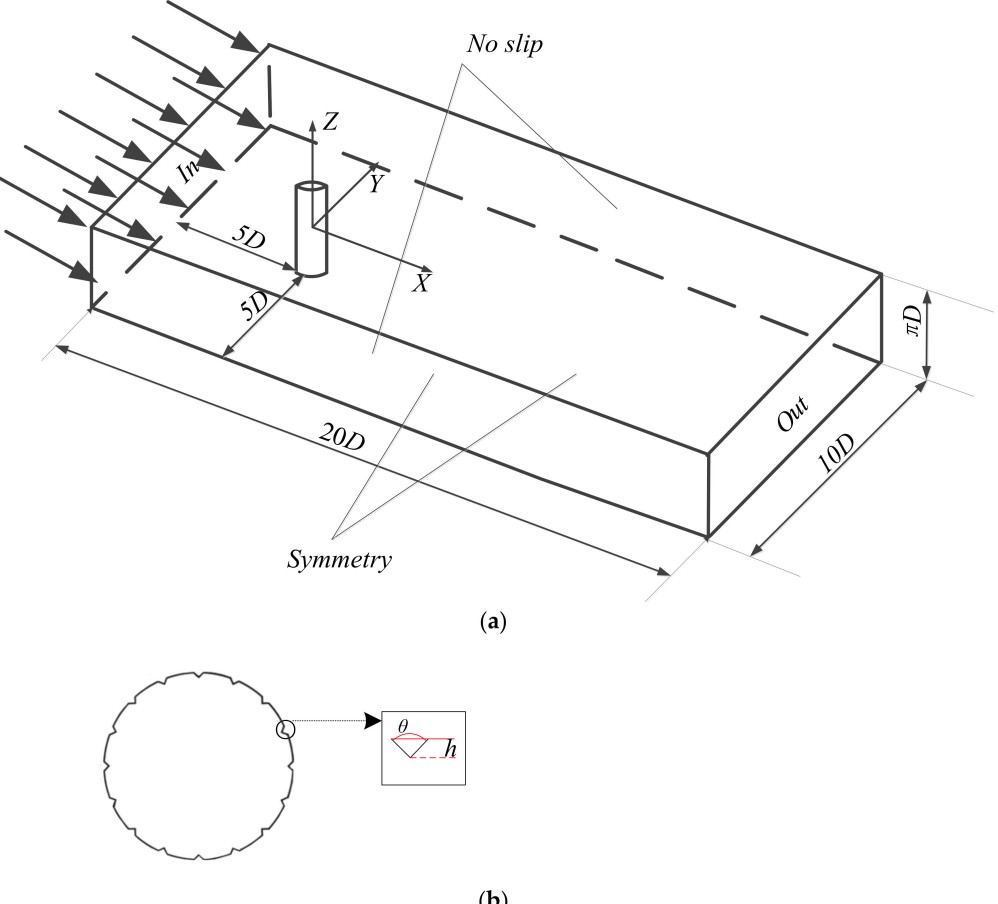

(a)

(b)

**Figure 2.** Computational domain and arrangement of circular cylinder. (**a**) Model of the fluid area and the coordinate system; (**b**) Schematic diagram of V-groove structure.

The calculation inlet is set to the velocity inlet ($U_\infty$ = 1.5 m/s), the outlet is a pressure outlet, the upper and lower sides are symmetry planes, and the left and right sides of the boundary have a no-slip wall surface. The pressure velocity coupling is solved by the SIMPLEC algorithm in a second-order upwind scheme. The specific simulation parameter settings are shown in Table 1. Figure 3 presents the schematic diagram of the computational domain grid. The O-type method is used to divide the grid and locally densify 35 layers of the grid.

**Table 1.** Simulation parameter settings.

| Parameter | Symbol | Unit | Numerical Value |
|---|---|---|---|
| Speed inlet | $U_\infty$ | m/s | 1.5 |
| Pressure outlet | $P$ | Pa | 1 |
| Time Step | - | s | $1 \times 10^{-3}$ |
| Kinematic viscosity number | $\nu$ | m$^2$/s | $1.01 \times 10^{-6}$ |
| density | $\rho$ | kg/m$^3$ | 1025 |
| Residual accuracy | - | - | $1.0 \times 10^{-5}$ |
| Cylindrical diameter | D | mm | 20 |

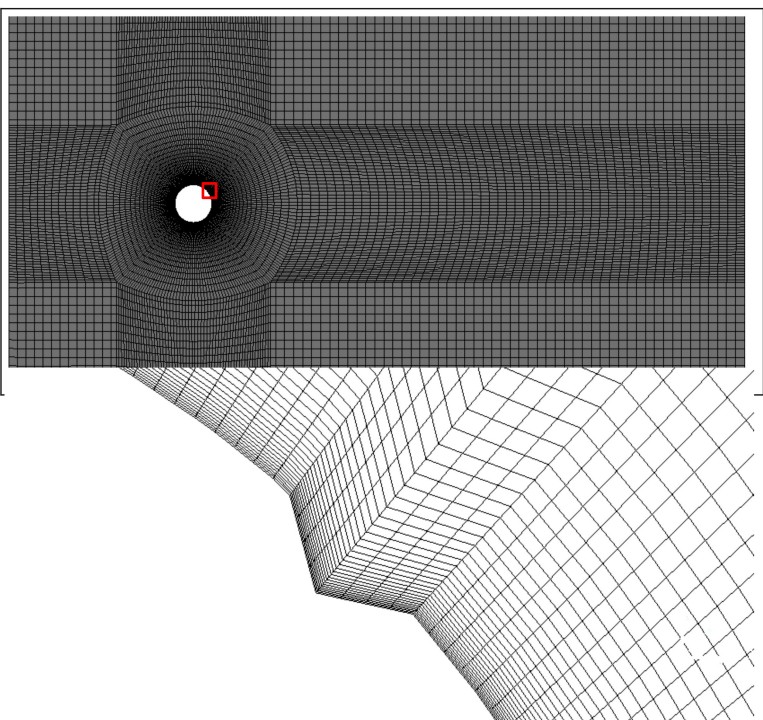

**Figure 3.** The schematic diagram of the computational domain structured grid.

### 2.4. Experimental Equipment

Particle image velocimetry (PIV) technology was first proposed by Adrian [18] in 1984. After nearly 40 years of development, PIV technology has made great progress in the aspects of dimension calculation, precision calculation, and real-time performance. In recent years, it has been developing in terms of stereoscopic, whole-system, chromatography, and micro-PIV, which can well meet the needs of people's analysis of fluid phenomena.

The PIV system is built on a circulating water tank, which is about 4 m long with a rectangular cross-section of 0.3 m × 0.3 m (width × height). As shown in Figure 4, the PIV system is mainly composed of a light source, high-speed camera, tracer particles, and image processing system. The light source comes from a laser, which can produce a continuous light source with a thickness of 1 mm during the experiment.

A high-speed camera (pco. dimax S1, Excelitas Technologies®Corp, Munich, Germany) with a resolution of 1008 × 1008 pixels was used to capture the 2000 successive digital particle images at a time interval of 1 ms (i.e., 1000 frame per second) between two consecutive images and the shutter speed of each frame was set at 1.5 μs.

The tracer particles were well mixed in the fluid, and the diameter of the tracer particle was 10 μm, which satisfies the accuracy of the measurement.

The image processing system is the core of the PIV system. At present, the widely used method is the cross-correlation algorithm [19] based on Fourier transform to process the image of tracer particles and to receive the flow information of the whole fluid region.

Based on the conclusion of data analysis of the simulation, five different cases were designed for the experiment. The five cases are defined as follows: *case0* represents the smooth cylinder, while *case1, case2, case3* and *case4* respectively represent cylindrical surfaces with V-groove depths of $h/D = 0.02$, $h/D = 0.03$, $h/D = 0.04$, and $h/D = 0.05$. The cross-sections of the cylinders are shown in Figure 5. Furthermore, the length-to-diameter of all these cylinders is 15, which is large enough to make sure of a *2D* flow of the cylinder's near-wake [20].

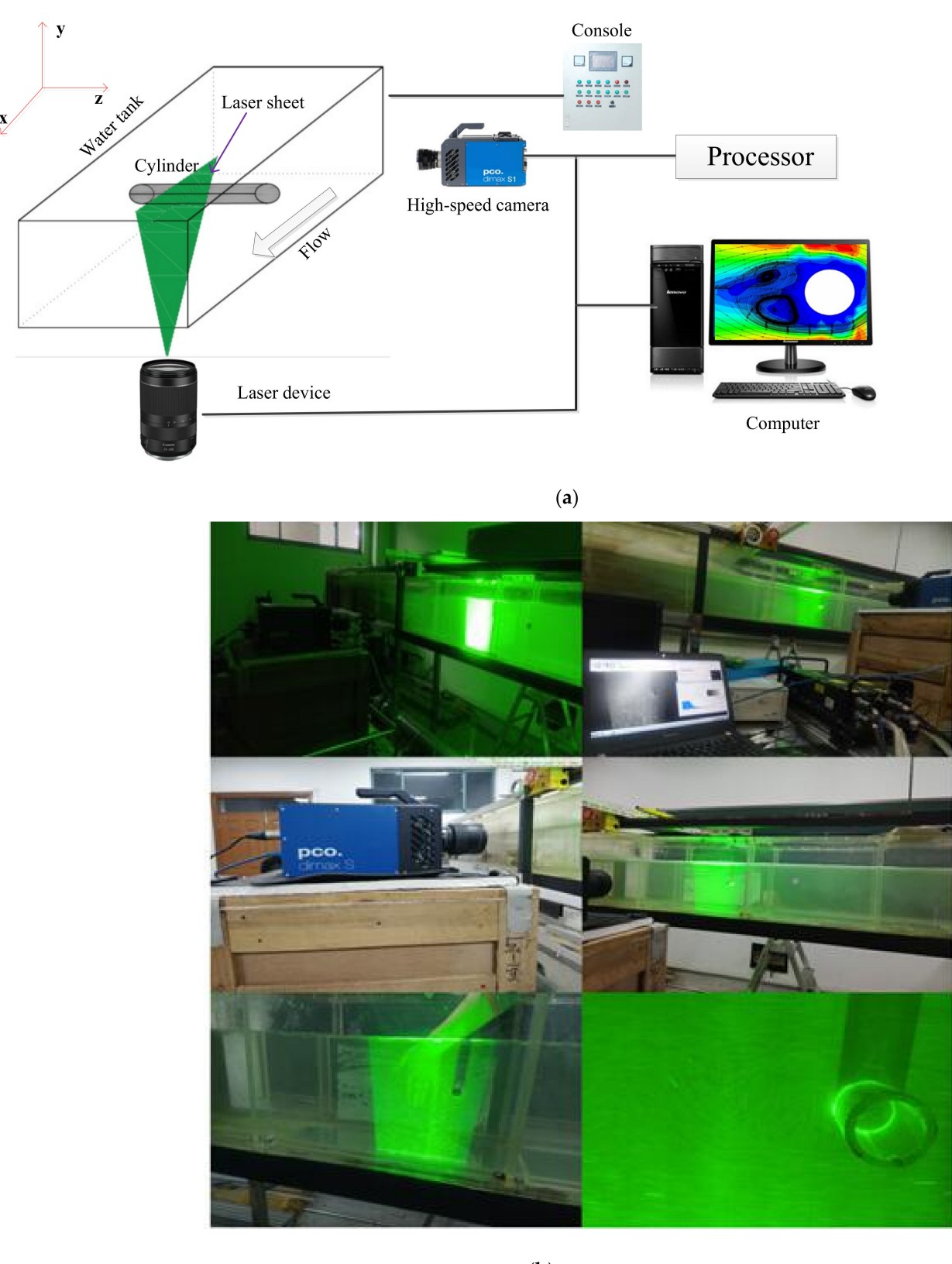

(**a**)

(**b**)

**Figure 4.** Experimental setup in the water tank. (**a**) Experimental schematic diagram. (**b**) Equipment diagram of experiment site.

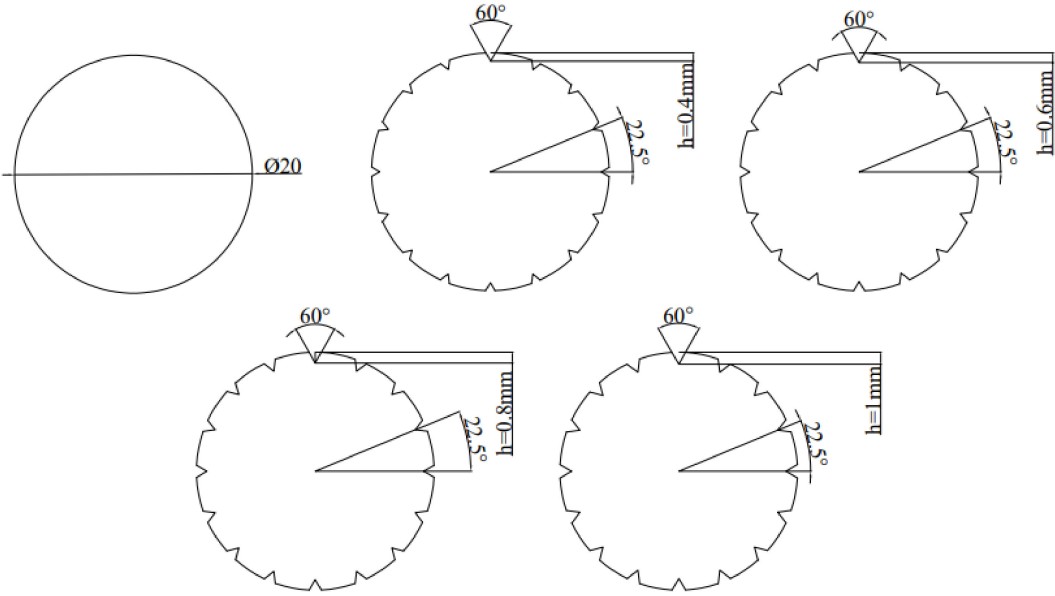

**Figure 5.** The schematic sketches of the cylinder cross-sections.

## 3. Model Verification

### 3.1. Grid-Independent Verification

The necessary grid-independent verification is carried out on the smooth cylinder when *Re* is $1.4 \times 10^4$. By locally densifying the cylinder's surrounding grids, the grid numbers are about $5.0 \times 10^5$, $8.0 \times 10^5$, and $1.0 \times 10^6$, respectively. As shown in Figure 6, when the number of grids is small, the calculation error is significant. It can also be found that the values of $C_d$ (time-averaged drag coefficient) tend to be stable when the number of grids is $8.0 \times 10^5$, and the data differences between $8.0 \times 10^5$ and $1.0 \times 10^6$ are around 0.83%, indicating that a further increase of mesh resolution would have a negligible effect on the results of the numerical simulation. Therefore, the number of grids for $8.0 \times 10^5$ will be applied for calculation.

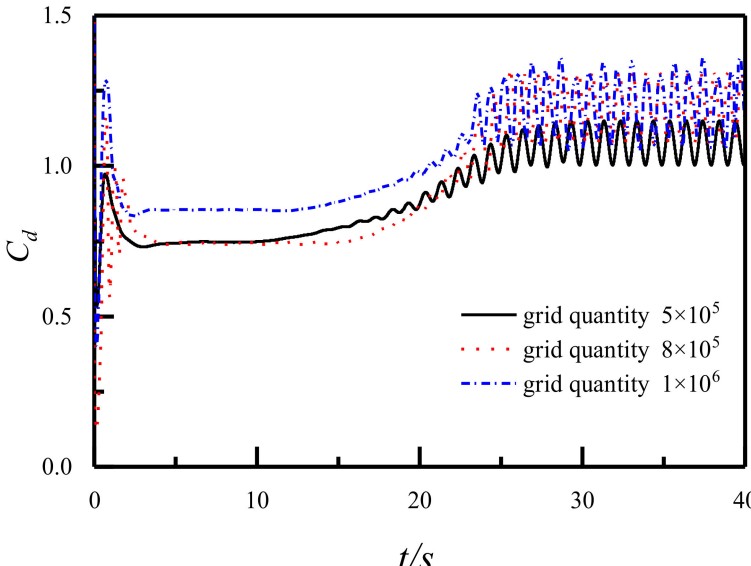

**Figure 6.** The grid verification.

Reynolds number:

$$\mathrm{Re} = \frac{\rho U_\infty D}{\mu} \tag{7}$$

Drag coefficient:

$$C_d = \frac{2Fd}{\rho U_\infty^2 A} \tag{8}$$

where $\rho$, $\mu$, $U_\infty$, and $D$ are the fluid density, the fluid viscosity, the flow velocity, and the diameter of the cylinder respectively, while $Fd$ is the drag of the cylinder and $A$ is the windward area.

### 3.2. Parameter Verification

After grid-independent verification, the smooth cylinder was selected for prediction under the condition of $Re = 1.4 \times 10^5$. Relevant parameters $C_d$, $C_l$, and $S_t$ are shown in Figure 7. Table 2 presents the data obtained by Schewe [21], Zdravkovich [22], and this paper. According to the comparison, the maximum error of $C_d$ and $St$ is 2.5%, the numerical simulation are very close to the experimental values, indicating that the numerical simulation methods are highly reliable in this paper.

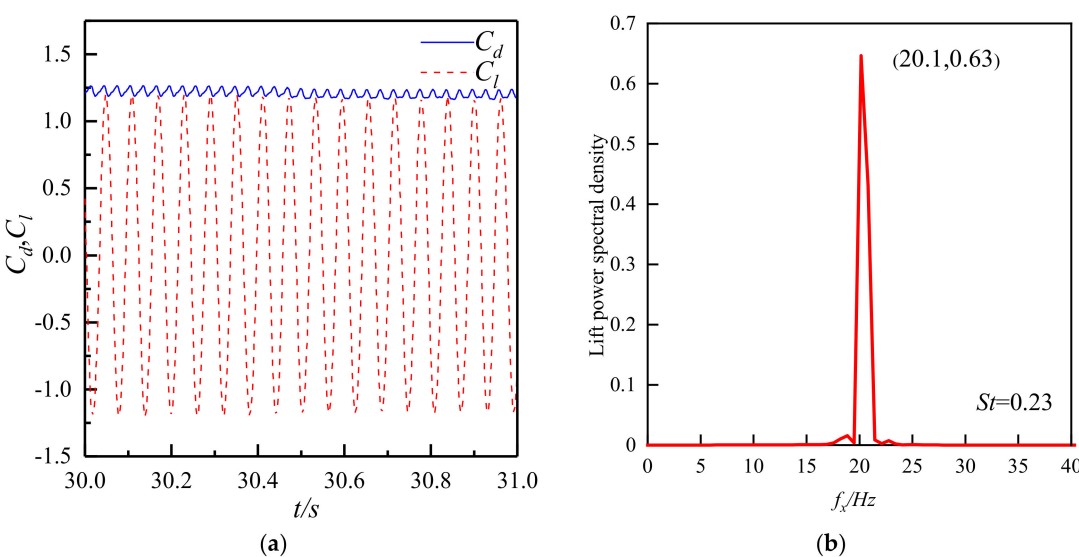

**Figure 7.** Numerical simulation result of Re = 14000; (**a**) drag coefficient, (**b**) lift power spectral density.

**Table 2.** Comparison of experimental data.

| Source of Literature | $C_d$ | $S_t$ | Experiment or Simulation |
|:---:|:---:|:---:|:---:|
| Schewe | 1.18 | 0.21 | experiment |
| Zdravkovich | 1.2 | 0.20 | experiment |
| This article | 1.21 | 0.23 | simulation |

Lift coefficient:

$$C_l = \frac{2Fl}{\rho U_\infty^2 A} \tag{9}$$

Strouhal number:

$$S_t = \frac{f_x D}{U_\infty} \tag{10}$$

where $Fl$ is the lift of the cylinder, $fx$ is the vortex shedding frequency.

The comparison of another important parameter, pressure coefficient ($C_p$), is shown in Figures 8 and 9, where $C_p$ was extracted at the cross section of the cylinder $z = \pi D/2$. As shown in Figure 8, $\alpha$ is the monitoring angle, with the initial monitoring angle at zero ($\alpha = 0°$). The monitoring point is evenly set at intervals of 20° along the downstream

direction from 0° to 180°. Figure 9 shows the distributions of $C_p$ acquired by Cantwall [23] and this paper, while the data errors can be ignored.

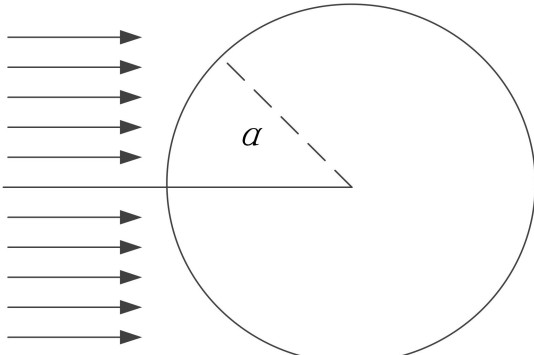

**Figure 8.** Schematic of the pressure monitoring point layout.

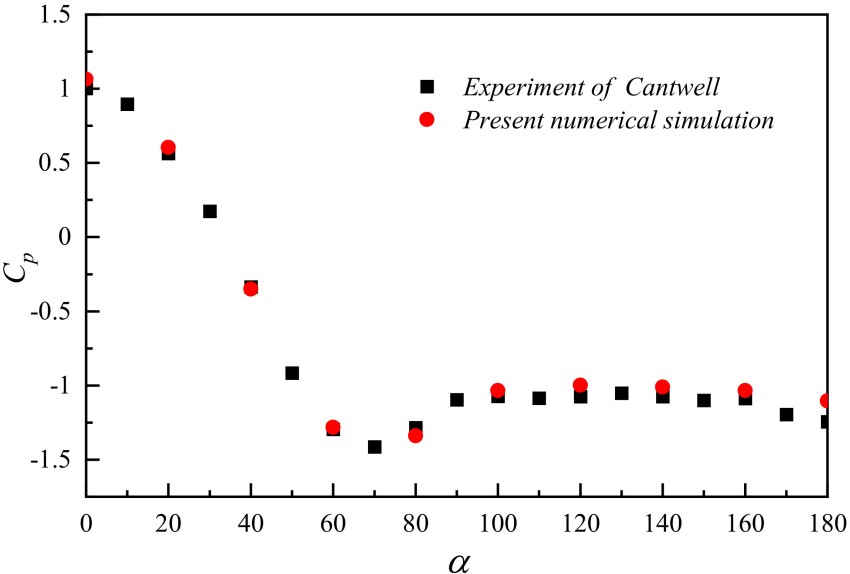

**Figure 9.** Pressure coefficient verification of the numerical simulation.

Pressure coefficient:

$$Cp = [p_{(\theta)} - p_{(\infty)}]/\frac{1}{2}\rho U_\infty{}^2 \tag{11}$$

where $p_{(\theta)}$ is the time-averaged pressure of different angles at the cylindrical surface and $p_{(\infty)}$ is the pressure at the inlet.

## 4. Numerical Simulation and Analysis

### 4.1. Lift Coefficient and Drag Coefficient

Figure 10 presents the profiles of $C_d$, $C_l$, and the power spectral density (PSD) for different V-groove depths. Comparing the PSD of lift for different V-groove depths, it can be seen that PSD has an obvious peak value for each case. The peak values of *case0*, *case1*, *case2*, *case3*, and *case4* decrease in turn, indicating that the vortex shedding is affected by the V-groove depth. For *case3* and *case4*, PSD presents a strong peak and a weak peak, indicating that the vortex shedding reaches bistability [24]. The double-peaks phenomenon is more obvious in *case4*, indicating that the law of vortex shedding is disturbed with the increasing V-groove depth, and the wake vortex becomes more complicated. The V-groove structure changes the alternate shedding state of the vortex, making the vortex shedding unstable.

Figure 11 depicts $C_d$ and the root mean square of $C_l$ ($C_{lrms}$) for different V-groove depths. With the increase of the V-groove depth, $Cd$ and $Cl_{rms}$ decrease first and then increase. For *case3*, the $C_d$ is 0.8 and $C_{lrms}$ is 0.73, and it reaches the lowest value in all cases. Meanwhile, $C_d$ and $C_{lrms}$ of *case3* decreases about 34% and 16% relative to *case0*, meaning that *case3* has the best drag reduction in all cases.

### 4.2. Pressure Coefficient

The study of Achenbach [25] drew a conclusion that drag mainly derives from pressure drag, which accounts for more than 98% of the full drag.

Figure 12 presents the distribution of $C_p$ around the cylinders with different V-groove depths. It can be seen that $C_p$, in the initial section of the cylinders (0° to 40°), does not change a lot, then the values of $C_p$ for all cases differ from each other for $\theta$ from 40° to 140°. Compared with *case0*, the pressure differences are obviously reduced between $\theta = 0°$ and $\theta = 180°$ of *case1*, *case2*, *case3*, and *case4*, indicating that the cylinders with V-groove structure can well reduce the pressure drag. Among these, *case3* ($h/D = 0.04$) shows the smallest pressure difference, indicating that the drag reduction effect is the best. This phenomenon can be a good explanation for the result presented in Figure 11.

### 4.3. Time-Averaged Velocity of Flow Field

To analyze the influence of the different V-groove surface depths on velocity distribution, this paper selects the downstream velocity monitoring lines distribution, as shown in Figure 13. A plane perpendicular to the z-axis direction ($z = \pi D/2$) is taken from the cross-section of the cylinder. Four monitoring lines are set on this plane at $x/D = 0.8$, 1.4, 2.0 and 2.6, respectively. The length of the monitoring line is $5D$, and more than 40 monitoring points are collected on each monitoring line.

Figure 14 shows the time-averaged downstream velocity ($U/U_\infty$) distributions for different cases. It can be seen that $U/U_\infty$, along the y-direction from $-2.5D$ to $2.5D$, is affected by the cylinder at these locations ($x/D = 0.8$, 1.4, 2.0 and 2.6). As the V-groove depth increases, the width of the wake flow field gradually decreases, and the reduction rate of the wake flow field of *case4* is about 50% of that of *case0*. The reason might be due to the fact that the absorption of the V-grooves delays the detachment of the vortex, and the shedding position is closer to the tail edge of the cylinder.

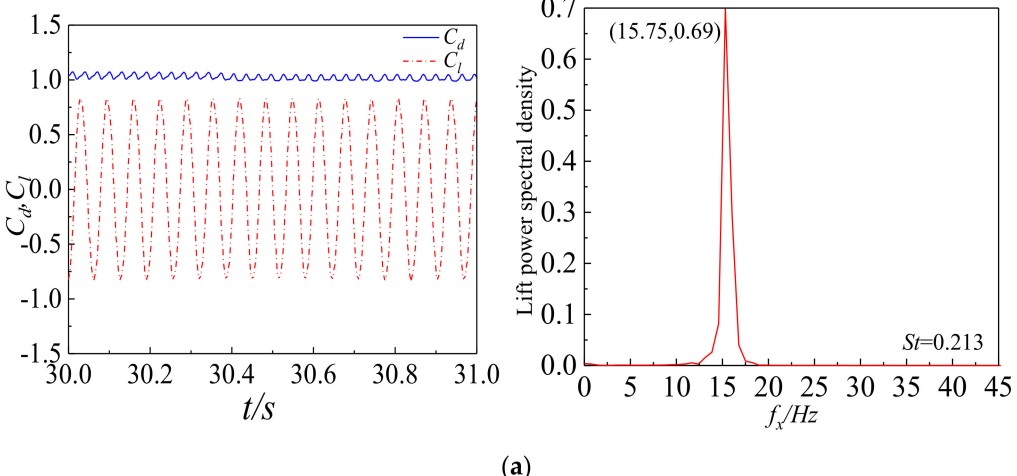

(**a**)

**Figure 10.** *Cont.*

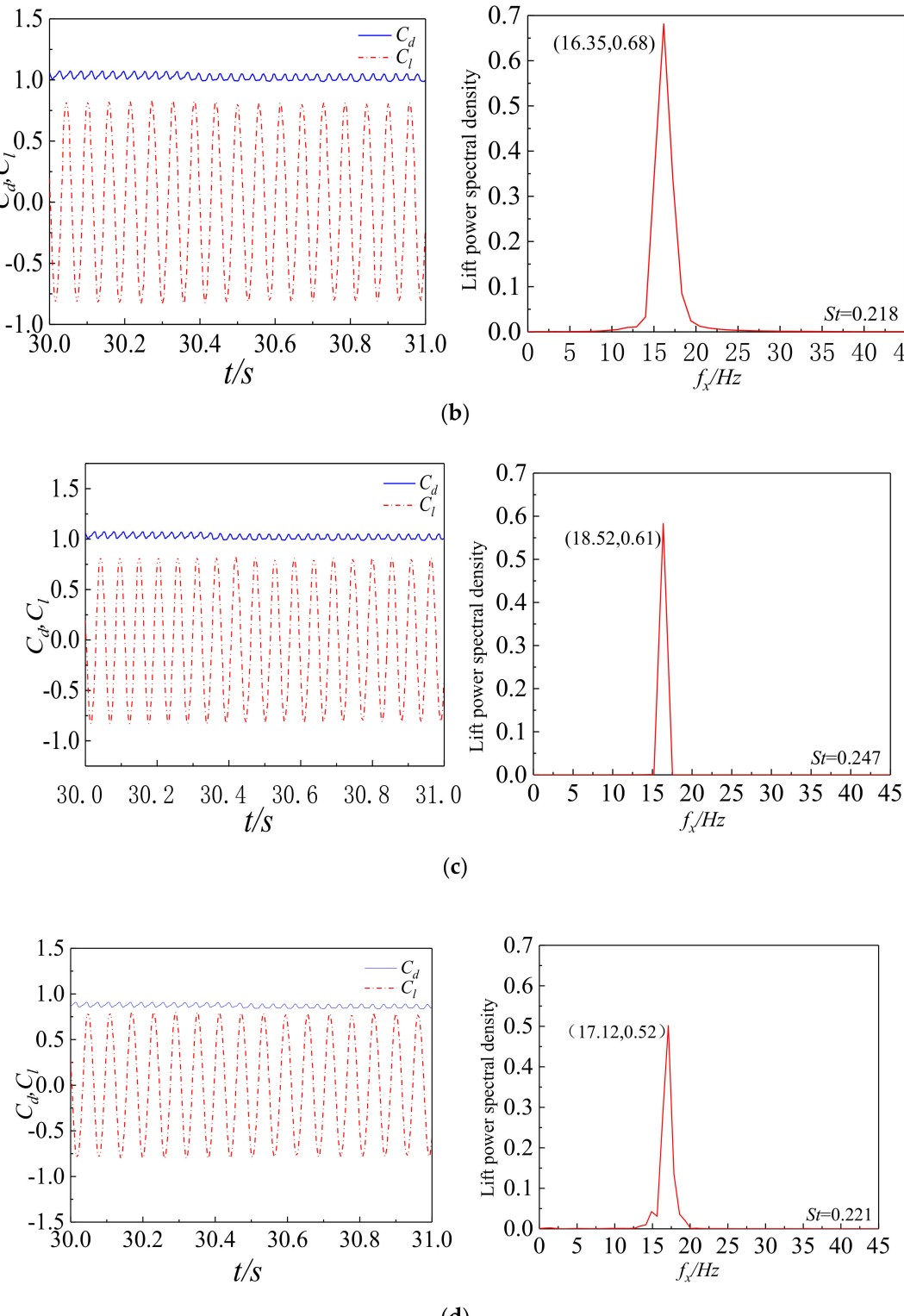

**Figure 10.** *Cont.*

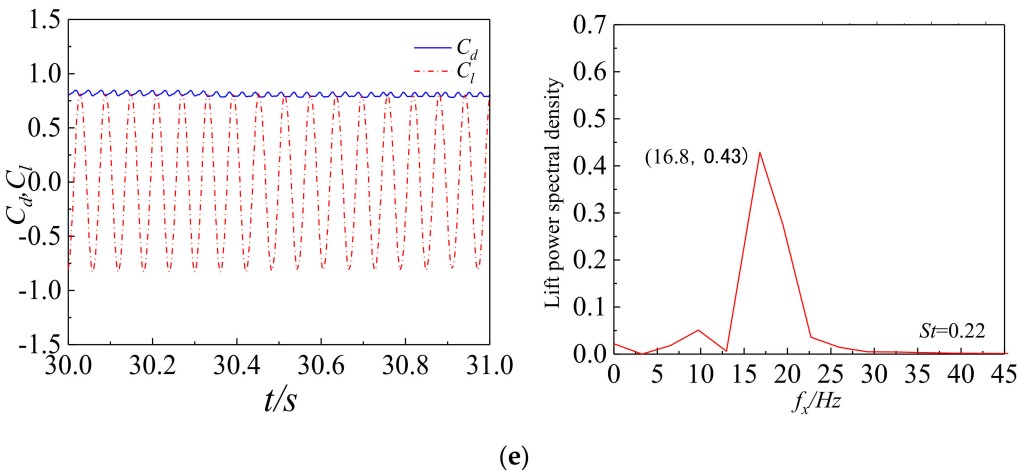

(**e**)

**Figure 10.** Profiles of drag coefficient, lift coefficient and vortex shedding frequencies for different V-groove depths; (**a**) *Case0*; (**b**) *Case1*; (**c**) *Case2*; (**d**) *Case3*; (**e**) *Case4*.

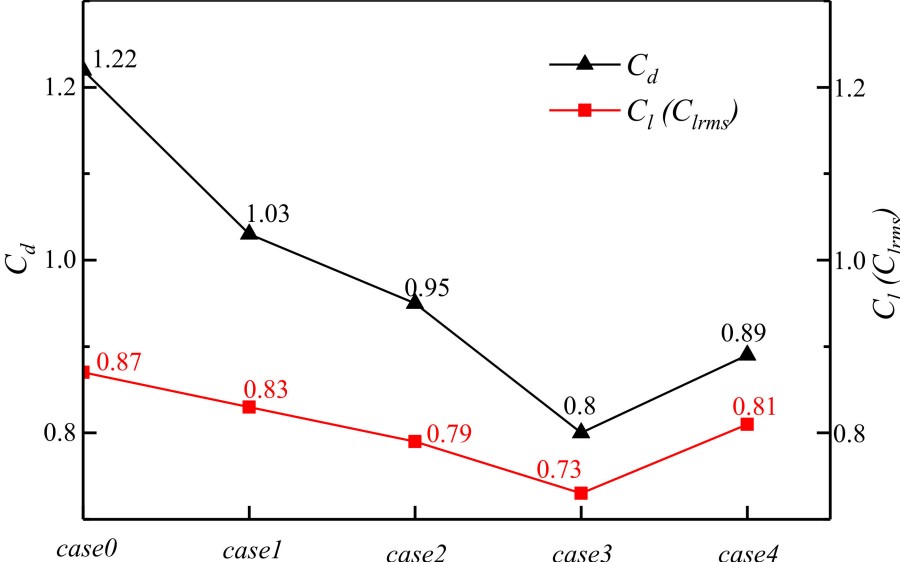

**Figure 11.** Mean drag coefficient and lift coefficient mean square error with different V-groove depths.

At the position of $x/D = 0.8$ (Figure 14a), it is interesting to note that the $U/U_\infty$ at the center for *case0* is negative, indicating that there is velocity backflow in the wake flow field. However, this phenomenon does not exist in *case1*, *case2*, *case3*, and *case4*, demonstrating that the V-groove changes the velocity distribution and eliminates velocity backflow. Moreover, with the increase of V-groove depth, the control effect on $U/U_\infty$ is more obvious, and the control efficiency reaches about 80% for *case4* compared with *case0*.

At the position of $x/D = 1.4$ and $x/D = 2.0$, as shown in Figure 14b,c, and different from the position of $x/D = 0.8$, the widths of the wake flow field are decreased. On the other hand, $U/U_\infty$ gradually increases when the monitoring distance is far from the center of the cylinder. This is because the influence of the cylinder on the wake flow velocity decreases with the increase of monitoring distance.

At the position $x/D = 2.6$, as presented in Figure 14d, there is almost no difference in $U/U_\infty$ between the smooth cylinder and the cylinder with the different V-groove depths from $y = -2.5D$ to $2.5D$, indicating that the V-groove surface mainly affects the $U/U_\infty$ of the wake flow field from $x = 0$ to $2.6D$. Due to the development of the wake flow vortex, it can be found that the $U/U_\infty$ of *case3* is greater than that of other cases at the position $x/D = 1.4$,

2.0, and 2.6. This result indicates that *case3* can most effectively reduce the velocity and the pressure difference, as per the conclusion plotted in Figure 12.

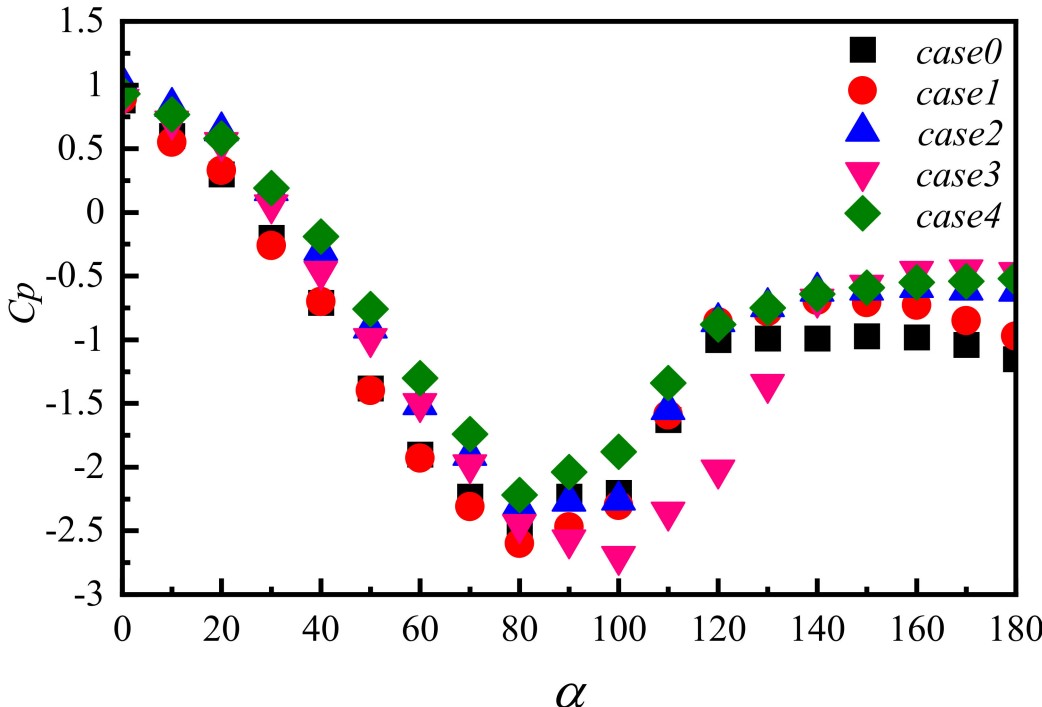

**Figure 12.** Distribution of pressure coefficient around circular cylinder for different V-groove depths.

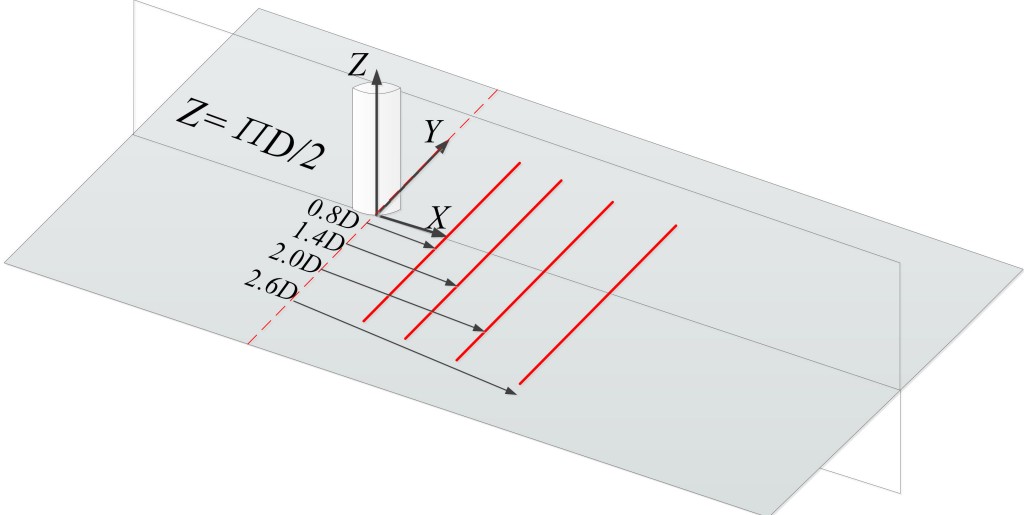

**Figure 13.** Downstream velocity monitoring line layout.

It can be seen from the time-averaged velocity diagram that the flow in the wake area changes. The flow velocity of the fluid in the wake area of a smooth cylinder is much lower than that of the grooved cylinder. The reason may be that the large number of secondary vortices are formed in the groove structure, which makes the contact between the fluid and the solid change into contact between fluid and fluid and reducing frictional resistance; thus the velocity of the smooth cylindrical flow field would decrease faster than that of the grooved cylindrical flow field. It can be concluded that the groove structure can reduce the degree of fluid flow velocity drop in the flow field.

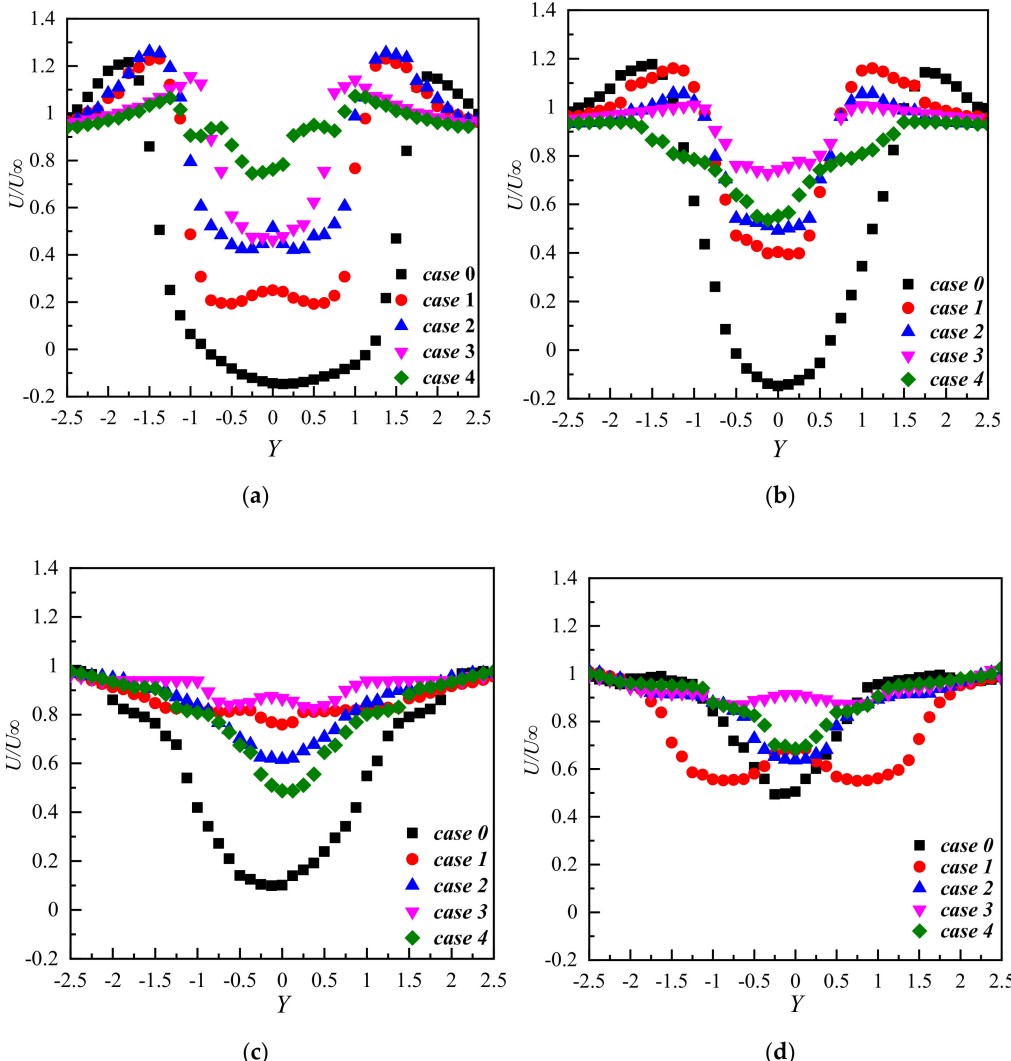

**Figure 14.** Time-average velocity distribution of different V-groove depths under different *x/D* conditions; (**a**) *x/D* = 0.8; (**b**) *x/D* = 1.4; (**c**) *x/D* = 2.0; (**d**) *x/D* = 2.6.

## 5. Experimental Results and Discussions

### 5.1. Velocity Field

To verify the conclusion drawn from the simulation data, the time-average velocity under $Re = 6.8 \times 10^3$ and $Re = 3 \times 10^4$ are extracted from the experimental flow field, as shown in Figure 15a,b. It can be found from Figure 15a,b that the curve shows a "U" shape near the cylinder, while the curve changes and presents a "V" shape away from the cylinder [26], something which is also reflected in the simulation data. However, the values of velocity in experiments are generally lower than those in simulation, which may be due to the mixing of tracer particles and other impurities in the fluid in experiments.

For *case3*, its velocity recovery rate is the fastest at $Re = 3 \times 10^4$, meaning that the pressure difference of the V-groove cylinder with *h/D* = 0.04 is also reduced in the experiment, as shown in Figure 15b. However, for *case4*, the velocity recovery rate is the fastest at $Re = 6.8 \times 10^3$, as shown in Figure 15a. This may be because when the Reynolds number is less than $3 \times 10^4$, the V-groove cylinder with *h/D* = 0.05 has a relatively large influence on the velocity [27]. Also, the width of the wake flow for the smooth cylinder is larger than the cylinder with the different V-groove surface depths. In addition, the results of the numerical simulation show that the velocity minimum value gradually increases with the increase of distance. At the location of *x/D* = 0.8 (Figure 14a), the velocity minimum value of *case1* is −0.18, while the velocity minimum value of the V-groove cylinder is greater than 0.2.

As *x/D* increases to 2.6, the velocity minimum value of *case1* reaches 0.5 with a weakened wake influence, while the velocity minimum value of other cases also increases. Figure 15b shows the velocity change trend of the experiment, At the location of *x/D* = 0.8, the velocity minimum value of the smooth cylinder is close to 0, and the velocity minimum value of the V-grooved cylinder is 0.027–0.086. As *x/D* increases to 2.6, the velocity minimum value changes significantly, the smooth cylinder reaches 0.28, and the V-groove cylinder reaches 0.35–0.42, respectively. There are numerical differences between the results of the experiment and the simulation. This is due to errors and interference in the actual experiment, which cannot reach the accuracy of the simulation calculation, but the experimental results show the same trend as the numerical simulation. Therefore, the conclusion obtained from the experimental data is consistent with the numerical simulation.

It can be seen from the comparison of velocity diagrams that the calculation model selected in this paper has high accuracy.

*5.2. Vortex Structure*

To provide a more intuitive analysis of the influence that different V-groove surface depths have on the vortex shedding at the near-wall surface, a plane perpendicular to the z-axis direction is taken from a cross-section of the cylinder. Then the features of the vortex shedding for all cases in this section are plotted in Figures 16 and 17.

Figure 16 describes the streamlines near the cylinder with different V-groove depths at $Re = 6.8 \times 10^3$. It can be seen that the vortex structure presents a complete symmetry for case 0, as demonstrated in Figure 16. For c*ase1*, *case2*, *case3*, and *case4*, the symmetry becomes insignificant, this is probably because small vortexes are generated at the groove of the cylinders [27,28], which will interfere with the development of the large vortices and destroy the stability of the periodic changes in the wake region. This is also one of the important reasons for the drag reduction effect of the cylinder with the V-groove surface. This is consistent with the phenomenon of the change in drag coefficient for the cylinder with the V-groove surface. A similar phenomenon can also be seen in Figure 17, at $Re = 3 \times 10^4$, indicating that the two Reynolds numbers have little effect on the vortex structure.

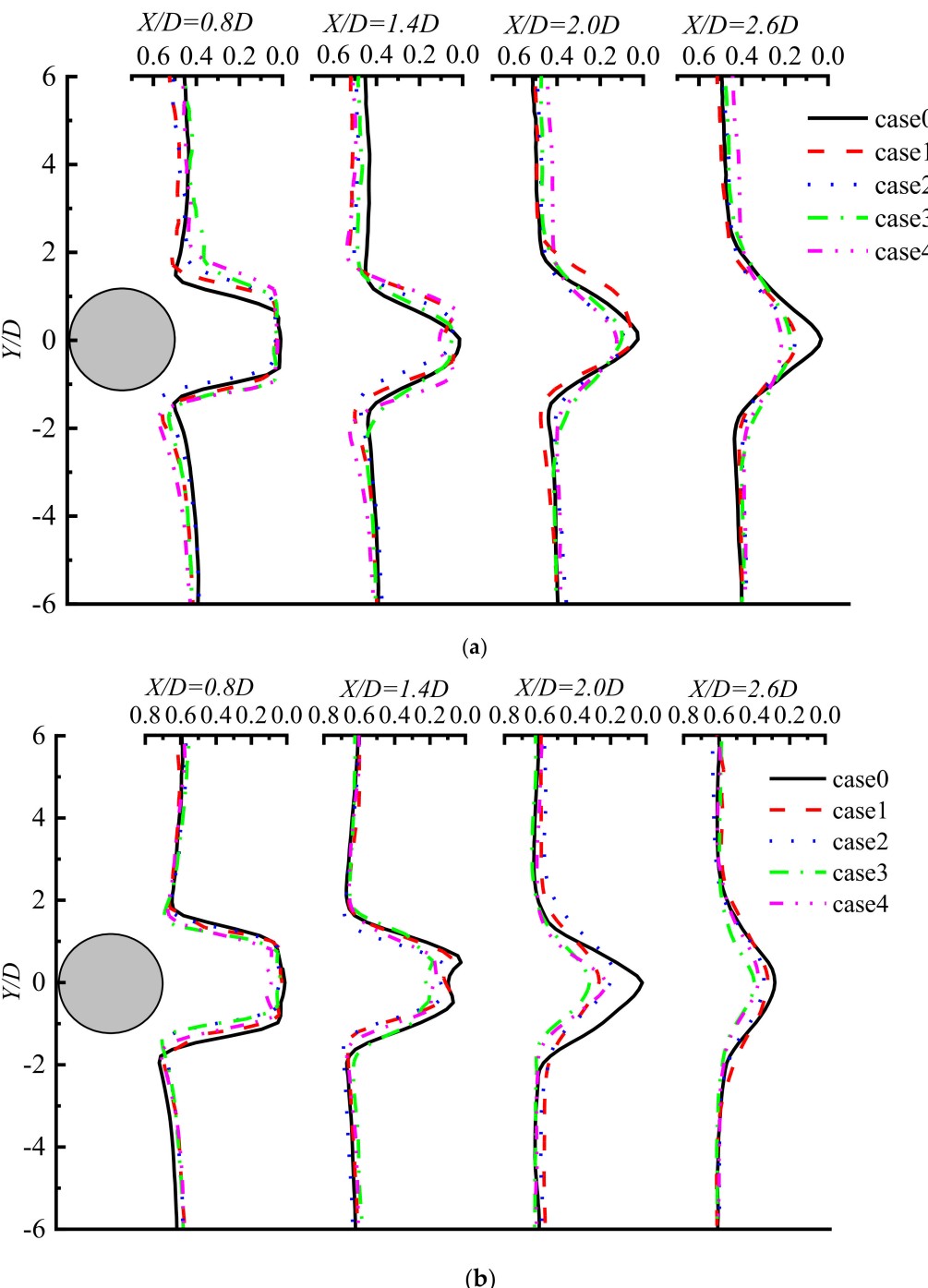

**Figure 15.** Mean streamwise velocity distribution of the experiment at different positions with different V-groove depths; (**a**) $Re = 6.8 \times 10^3$; (**b**) $Re = 3 \times 10^4$.

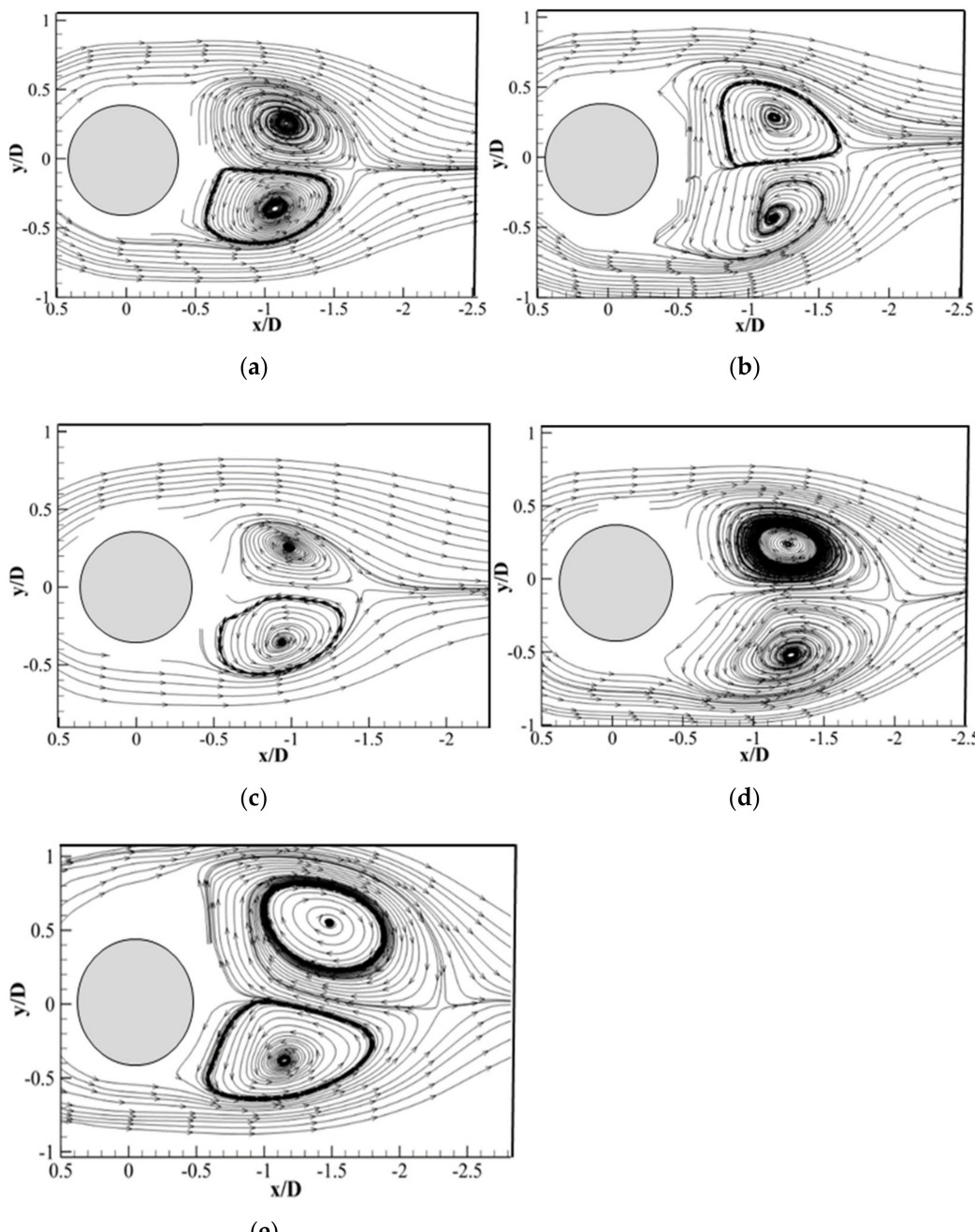

**Figure 16.** The streamlines near the circular cylinder with different V-groove depths at *Re* = 6.8 × 10³; (**a**) *case0*; (**b**) *case1*; (**c**) *case2*; (**d**) *case3*; (**e**) *case4*.

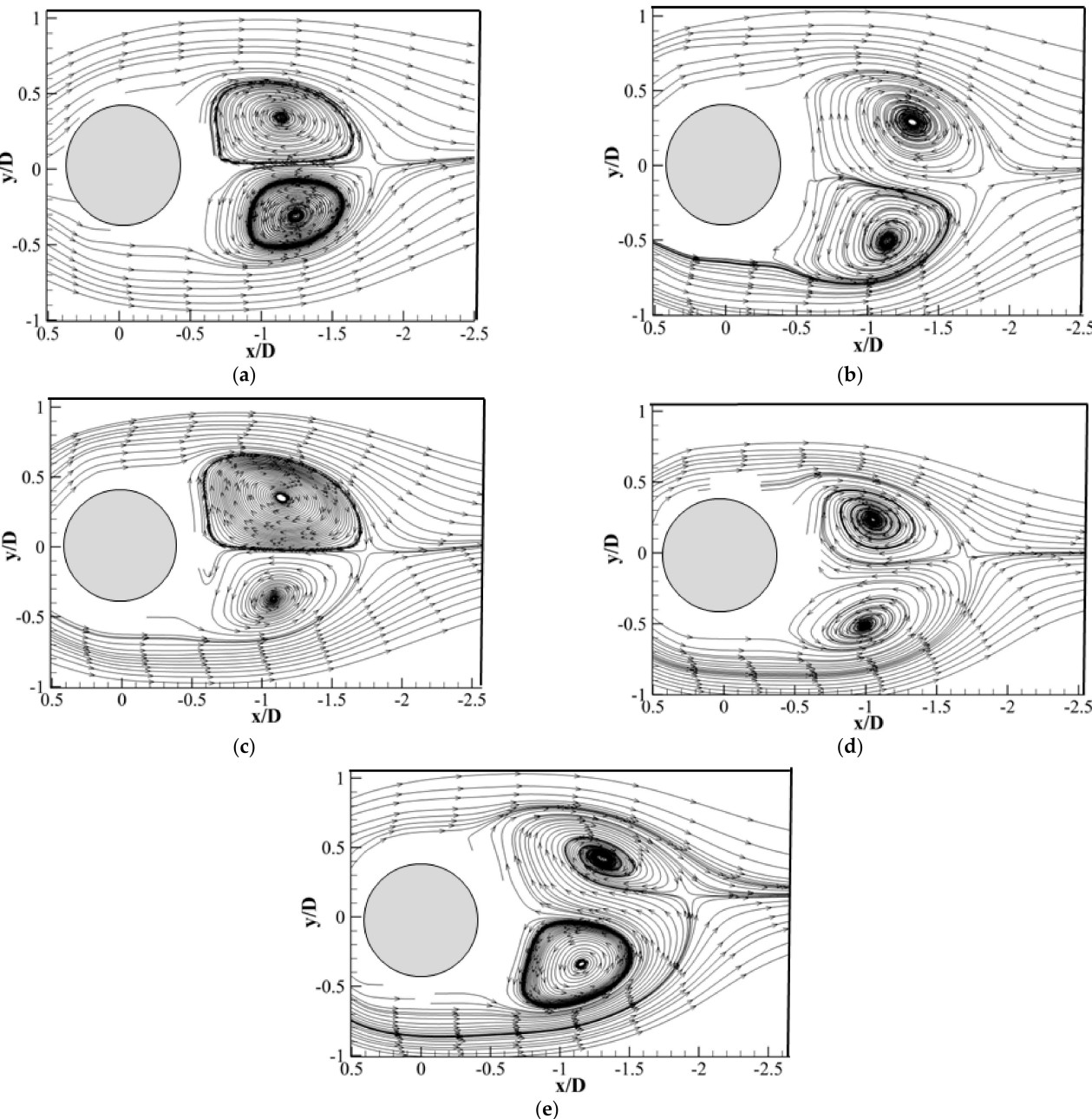

**Figure 17.** The streamlines near the circular cylinder with different V-groove depths at *Re* = 3 × 10$^4$; (**a**) *case0*; (**b**) *case1*; (**c**) *case2*; (**d**) *case3*; (**e**) *case4*.

## 6. Conclusions

In this study, a V-shaped groove surface with different depths is arranged on a cylinder surface, while a CFD *k-ω/SST* turbulence model and particle image velocimetry (PIV) technology are employed to study the drag reduction effect and wake characteristics. The following conclusions can be drawn:

(1) The vortex shedding frequency is changed by the V-groove structure. With the increase of V-groove depth, Strouhal number shows a trend of increasing first and then decreasing.

(2) The drag coefficient and the pressure difference between the head and tail are reduced for the cylinder with the different V-groove surface depths, a phenomenon that is more obvious when *h/D* = 0.04.

(3) With an increase of V-groove depth, the width of the wake flow field gradually decreases, and the reduction rate of the wake flow field of the V-groove structure is less than that of the smooth cylinder, *case4* is about 50% of that of *case0*. The experimental results are consistent with the simulation results.

(4) The release process of the vortex is disturbed by the V-groove surface. And the V-groove surface shatters the larger vortexes and turns them into a larger number of small vortexes.

(5) The vortex structure presents a complete symmetry for the smooth cylinder, however, the symmetry of the vortex structure becomes insignificant for the V-shaped groove structure with different depths.

**Author Contributions:** Conceptualization, J.Q.; methodology, J.Q.; validation, F.Y. and Y.Q.; formal analysis, J.Q.; investigation, F.Y.; data curation, Y.Q.; writing—original draft preparation, J.Q.; writing—review and editing, F.Y.; visualization, Q.C. All authors have read and agreed to the published version of the manuscript.

**Funding:** This research received no external funding.

**Institutional Review Board Statement:** The authors choose to exclude this statement.

**Informed Consent Statement:** Not applicable.

**Data Availability Statement:** No new data were created or analyzed in this study. Data sharing is not applicable to this article.

**Acknowledgments:** The author wishes to acknowledge support given to him by Natural Science Foundation of Jiangsu Province of China (No.BK20191459), and National Natural Science Foundation of China (No.12002138).

**Conflicts of Interest:** The authors declare no conflict of interest.

## Nomenclature

| | |
|---|---|
| $D$ | the diameter of the cylinder, mm. |
| $h$ | the depth of groove, mm. |
| $U\infty$ | the speed inlet, m/s. |
| $P$ | the pressure outlet, pa. |
| $\rho$ | the fluid density. |
| $\mu$ | the fluid viscosity. |
| $F_d$ | the drag of the cylinder |
| $A$ | the windward area. |
| $F_l$ | the lift of the cylinder. |
| $f_x$ | the vortex shedding frequency. |
| $C_d$ | the drag coefficient. |
| $C_l$ | the lift coefficient. |

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
