# Peer review of "A Study of Drag Reduction on Cylinders with Different V-Groove Depths on the Surface"

_water, doi:10.3390/w14010036_

Round 1

Reviewer 1 Report

A study of drag reduction on cylinders with different V-groove depths on the surface

The paper studies the effect of the V-groove at the cylinder surface on the flow field by numerical and experimental approaches. 
I can evaluate both the approaches are made. However, discussion is very poor. Moreover, there are many careless misses. 
Accordingly I should reject the paper. 

First, the authors does not discuss why is the groove effective for the flow around cylinder. 
The numerical simulation and experiment may give many data, e.g., instantaneous flow field and statistics. Especially the contribution of the paper is the effect of the v-groove and the reader may have interest the flow around the groove. However, the authors do not display...

Second, the comparison between the numerical simulation and experimental date is few. In general, both approaches work complementary. E.g., Figures 14 and 15 can be made direct comparison, but the authors do not. 

Third, fundamental discussion is lacked. Not only the mean value, the rms values of the flow filed are also important. In addition, in Figures 16 and 17, the flow field has temporal variation, but the figures do not care about it. The authors consider the Strouhal number for the simulation, but does not in the experiment. I, therefore, think that it is difficult to get new insight from them. 

Minors are follows. 

Figure 12, how does the pressure drag measure? The cylinder surface has groove and it is difficult to measure the pressure at the same location over the groove. 

Figure 6, I cannot understand why the authors claim that there is not grid dependency. Please clarify the detail how do the author judge it. 
And the vertical label of Cd should be C_d (d is the subscript) and the author should follow the same manner for other labels.

Line 212, 1.4 \times 104, should be 10^4

Figure 2 is broken.

All the indices to denote variables and coordinates should be italic, e.g., D, X, Y etc. 

Author Response

   We are very grateful for this review’s detailed comments that have helped us improve significantly this manuscript. The major changes to the manuscript have been highlighted in red. Following are actions we have taken to address specific issues raised by the reviewer.

Reviewer’s comments:

1.First, the authors does not discuss why is the groove effective for the flow around cylinder. 
The numerical simulation and experiment may give many data, e.g., instantaneous flow field and statistics. Especially the contribution of the paper is the effect of the v-groove and the reader may have interest the flow around the groove. However, the authors do not display...

Author’s response:

This paper mainly studies the difference of drag reduction performance of cylinder with different V-grooves, the most direct way is to measure the lift coefficient and the drag coefficient, the velocity distribution of flow field, pressure coefficient. The vortex shedding and the change of vortex structure were studied experimentally. The flow around the groove is negligible for this paper.

Reviewer’s comments:

  1. Second, the comparison between the numerical simulation and experimental date is few. In general, both approaches work complementary. E.g., Figures 14 and 15 can be made direct comparison, but the authors do not. 

Author’s response:

Article 5.1 explains the experimental values and compares them with simulation data.

Reviewer’s comments:

3.Third, fundamental discussion is lacked. Not only the mean value, the rms values of the flow filed are also important. In addition, in Figures 16 and 17, the flow field has temporal variation, but the figures do not care about it. The authors consider the Strouhal number for the simulation, but does not in the experiment. I, therefore, think that it is difficult to get new insight from them. 

  Author’s response:

  Figs. 16 and 17 are supplements to the previous ones, aiming to show that the V-groove cylinder has the effect of drag reduction, and supplement to show the correctness of the resistance curve.

Reviewer’s comments:

4.Figure 12, how does the pressure drag measure? The cylinder surface has groove and it is difficult to measure the pressure at the same location over the groove. 

Author’s response:

Figure 12 is the simulation result. In Fluent software, pressure monitoring points can be set, pressure can be calculated and pressure curve can be drawn.

Reviewer’s comments:

5.Figure 6, I cannot understand why the authors claim that there is not grid dependency. Please clarify the detail how do the author judge it. 

Author’s response:

Fig. 6 shows that the influence of further improvement of grid resolution on numerical simulation results can be ignored.

And the vertical label of Cd should be Cd (d is the subscript) and the author should follow the same manner for other labels.

Author’s response:

Revised.

Reviewer’s comments:

6.Line 212, 1.4 \times 104, should be 10^4

Author’s response:

Revised.

Reviewer’s comments:

7.Figure 2 is broken.

All the indices to denote variables and coordinates should be italic, e.g., D, X, Y etc. 

Author’s response:

Revised.

Reviewer 2 Report

I have the following comments:

- the results are poorly presented. For some reason, experimental and numerical results are shown in separate section

- figure design should be revisited. In some places, for example, the flow goes from left to right, in some places it is the opposite. See fig 15, 16, for instance

Author Response

We are very grateful for this review’s detailed comments that have helped us improve significantly this manuscript. The major changes to the manuscript have been highlighted in green. Following are actions we have taken to address specific issues raised by the reviewer.

Reviewer’s comments:

1.The results are poorly presented. For some reason, experimental and numerical results are shown in separate section.

Author’s response:

The results have been supplemented.

Reviewer’s comments:

2.Figure design should be revisited. In some places, for example, the flow goes from left to right, in some places it is the opposite. See fig 15, 16, for instance.

Author’s response:

Revised.

Reviewer 3 Report

1) Due to a large number of abbreviations, symbols and markings used throughout the paper text, it will be very helpful to any reader that all of the mentioned is presented in one place. Therefore, adding a Nomenclature in which all of the mentioned elements will be listed and explained is required.

2) Figure 2 (both parts of this Figure) is missing. Please, provide complete Figure 2 – only then this Figure can be properly evaluated.

3) Figure 3 – both Figure parts should be enlarged to ensure better visibility. In my opinion, the best will be to present both parts of this Figure one under another, not in the same line as it is at the moment.

4) Along with Experimental setup scheme (Figure 4), please add a picture of the complete Experimental setup with marked main measuring equipment elements. This will be fully convincing evidence that the measurements are really performed (at the moment, the reader cannot be sure by seeing only the experimental scheme that the measurements are really performed).

5) Figure 5 – case 2 and case 3 – h is not 0.06 mm and 0.08 mm – the correct values are 0.6 mm and 0.8 mm. Therefore, Figure 5 should be corrected.

6) Figure 6 – in the paper text should be specified for which cylinder (case) of five observed, the grid-dependency verification is performed. Grid-dependency verification should be performed for at least one more cylinder.

7) Section 3. 2 – If it is possible, the verification of Parameters should be performed for at least two or more cylinders. If this process will be too complex or too much time-consuming, then this comment can be neglected.

8) Figure 10 – right part of each presented Figure (abscissa) – frequency is in Hz, not HZ.

9) Figure 11 – abscissa – h should be removed (before this Figure is explained which depth is related to which case).

10) Figure 15 – markings (a) and (b) should occur once in each Figure (at the moment, in this Figure they occur twice).

11) Conclusions – in this section should be involved more exact results obtained in the performed analysis (exact values).

12) Used References are dominantly older than 5-10 years. Please, involve much more recent literature from this research field in the paper text.

Author Response

We are very grateful for this review’s detailed comments that have helped us improve significantly this manuscript. The major changes to the manuscript have been highlighted in blue. Following are actions we have taken to address specific issues raised by the reviewer.

Reviewer’s comments:

  • Due to a large number of abbreviations, symbols and markings used throughout the paper text, it will be very helpful to any reader that all of the mentioned is presented in one place. Therefore, adding a Nomenclature in which all of the mentioned elements will be listed and explained is required.

Author’s response:

Revised.

Reviewer’s comments:

2) Figure 2 (both parts of this Figure) is missing. Please, provide complete Figure 2 – only then this Figure can be properly evaluated.

Author’s response:

Different V-groove depth images are explained in Figure 5.

Reviewer’s comments:

3) Figure 3 – both Figure parts should be enlarged to ensure better visibility. In my opinion, the best will be to present both parts of this Figure one under another, not in the same line as it is at the moment.

Author’s response:

Revised.

Reviewer’s comments:

4) Along with Experimental setup scheme (Figure 4), please add a picture of the complete Experimental setup with marked main measuring equipment elements. This will be fully convincing evidence that the measurements are really performed (at the moment, the reader cannot be sure by seeing only the experimental scheme that the measurements are really performed).

Author’s response:

Pictures have been added to the article.

Reviewer’s comments:

5) Figure 5 – case 2 and case 3 – h is not 0.06 mm and 0.08 mm – the correct values are 0.6 mm and 0.8 mm. Therefore, Figure 5 should be corrected.

Author’s response:

Revised.

Reviewer’s comments:

6) Figure 6 – in the paper text should be specified for which cylinder (case) of five observed, the grid-dependency verification is performed. Grid-dependency verification should be performed for at least one more cylinder.

Author’s response:

Fig. 6 verifies the grid independence, verifies the three grid quantities and selects the best scheme

Reviewer’s comments:

7) Section 3. 2 – If it is possible, the verification of Parameters should be performed for at least two or more cylinders. If this process will be too complex or too much time-consuming, then this comment can be neglected.

Author’s response:

Due to space constraints, the parameter verification of redundant columns was not carried out.

Reviewer’s comments:

8) Figure 10 – right part of each presented Figure (abscissa) – frequency is in Hz, not HZ.

Author’s response:

Revised.

Reviewer’s comments:

9) Figure 11 – abscissa – h should be removed (before this Figure is explained which depth is related to which case).

Author’s response:

Revised.

Reviewer’s comments:

10) Figure 15 – markings (a) and (b) should occur once in each Figure (at the moment, in this Figure they occur twice).

Author’s response:

Revised.

Reviewer’s comments:

11) Conclusions – in this section should be involved more exact results obtained in the performed analysis (exact values).

Author’s response:

Revised.

Reviewer’s comments:

12) Used References are dominantly older than 5-10 years. Please, involve much more recent literature from this research field in the paper text.

Author’s response:

Relevant literature has been added.

Reviewer 4 Report

It is a valuable manuscript, which can be acceptyed after that a suitable revision has been done on it according to tyhe following points.

  1. The aim of the manuscript and its novelty are not adequately described.
  2. In equation 2 the adoption of the nabla symbol is not compatible with the indicial notation. Substitute ∇2 with ∂2/∂x2i.
  3. Define the turbulent kinetic energy kt.
  4. The turbulence model has to be indicated as the kt-ω model, as the turbulent kinetic energy is represented by the symbol kt.
  5. Define ω as the ratio ε/kt, turbulent dissipation on turbulent kinetic energy.
  6. In section 2.2. there is a serious flawness. Indeed in the kt-ω model appears the eddy viscosity μ, which should be defined in terms of kt and ω. But the authors state that it is set to the constant value μ=0.000062, making meaningless the whole turbulence model. If the eddy viscosity is constant, why to solve the kt-ω equations? The whole section must be corrected and rewritten.
  7.  It is stated that (rows 107-110): Navier-Stokes equations (N-S equations) are used to solve the turbulence problem which consumes huge computational resources. A practical way to avoid huge computation demand on turbulence modeling is the implementation of the RANS equations (Reynolds-averaged N–S equations). The statement is rather vague in the present form. It can be accepted if an explicit reference is done to the DNS.
  8. Give details on the numerical solver. Is it a in-house, a commercial or open source one?

Author Response

We are very grateful for this review’s detailed comments that have helped us improve significantly this manuscript. The major changes to the manuscript have been highlighted in pink. Following are actions we have taken to address specific issues raised by the reviewer.

 Reviewer’s comments:

  1. The aim of the manuscript and its novelty are not adequately described.

Author’s response:

The aim of the manuscript that the purpose of the manuscript is to study the drag reduction performance of v-groove cylinder.

Reviewer’s comments:

  1. In equation 2 the adoption of the nabla symbol is not compatible with the indicial notation. Substitute ∇2 with ∂2/∂x2i.

Author’s response:

Revised.

 Reviewer’s comments:

  1. Define the turbulent kinetic energy kt.

Author’s response:

The definition has been given in this paper

Reviewer’s comments:

  1. The turbulence model has to be indicated as the kt-ω model, as the turbulent kinetic energy is represented by the symbol kt.

Author’s response:

Revised.

Reviewer’s comments:

  1. Define ω as the ratio ε/kt, turbulent dissipation on turbulent kinetic energy.

Author’s response:

Revised.

Reviewer’s comments:

  1. In section 2.2. there is a serious flawness. Indeed in the kt-ω model appears the eddy viscosity μt , which should be defined in terms of kt and ω. But the authors state that it is set to the constant value μt =0.000062, making meaningless the whole turbulence model. If the eddy viscosity is constant, why to solve the kt-ω equations? The whole section must be corrected and rewritten.

Author’s response:

Revised.

Reviewer’s comments:

  1.  It is stated that (rows 107-110): Navier-Stokes equations (N-S equations) are used to solve the turbulence problem which consumes huge computational resources. A practical way to avoid huge computation demand on turbulence modeling is the implementation of the RANS equations (Reynolds-averaged N–S equations). The statement is rather vague in the present form. It can be accepted if an explicit reference is done to the DNS.

Author’s response:

Revised.

Reviewer’s comments:

  1. Give details on the numerical solver. Is it a in-house, a commercial or open source one?

Author’s response:

ANSYS 15.0, Customer # 503068

Round 2

Reviewer 1 Report

Reviewer’s comments:

1.First, the authors does not discuss why is the groove effective for the flow around cylinder.
The numerical simulation and experiment may give many data, e.g., instantaneous flow field and statistics. Especially the contribution of the paper is the effect of the v-groove and the reader may have interest the flow around the groove. However, the authors do not display...

Author’s response:

This paper mainly studies the difference of drag reduction performance of cylinder with different V-grooves, the most direct way is to measure the lift coefficient and the drag coefficient, the velocity distribution of flow field, pressure coefficient. The vortex shedding and the change of vortex structure were studied experimentally. The flow around the groove is negligible for this paper.

→ I believe that the flow behind the cylinder is changed since the flow round the groove works. If it is ignored or is not discussed, I do not think that the discussion is not enough. I think that the authors can discuss it by using simulation data... 

Author Response

We are very grateful for this review’s detailed comments that have helped us improve significantly this manuscript. The major changes to the manuscript have been highlighted in red. Following are actions we have taken to address specific issues raised by the reviewer.

1.First, the authors does not discuss why is the groove effective for the flow around cylinder. 
The numerical simulation and experiment may give many data, e.g., instantaneous flow field and statistics. Especially the contribution of the paper is the effect of the v-groove and the reader may have interest the flow around the groove. However, the authors do not display...

Author’s response:

This paper mainly studies the difference of drag reduction performance of cylinder with different V-grooves, the most direct way is to measure the lift coefficient and the drag coefficient, the velocity distribution of flow field, pressure coefficient. The vortex shedding and the change of vortex structure were studied experimentally. The flow around the groove is negligible for this paper. 

→ I believe that the flow behind the cylinder is changed since the flow round the groove works. If it is ignored or is not discussed, I do not think that the discussion is not enough. I think that the authors can discuss it by using simulation data... 

Author’s response:

The change of flow velocity in the flow field is supplemented.

Reviewer 3 Report

The Authors have properly answered and perform corrections/additions to some of my comments. A few of them are not properly addressed (or are not addressed at all).

  • My previous comment 1 – The Authors did not add a Nomenclature in the revised paper. Again, the Nomenclature is, in my opinion, required.
  • My previous comment 6 – From Figure 5 is clear that the Authors have observed 5 different cases. In the paper text related to Figure 6 is still missing an explanation – for which of five observed cases the grid verification is performed.
  • My previous comment 12 – A recent literature is not added. Only one reference is changed in comparison to the previous paper version. Used References are still dominantly older than 5-10 years. Therefore, a proper and recent literature addition is required.

From the above, the paper requires further revision.

Author Response

We are very grateful for this review’s detailed comments that have helped us improve significantly this manuscript. The major changes to the manuscript have been highlighted in green. Following are actions we have taken to address specific issues raised by the reviewer.

  1. My previous comment 1 – The Authors did not add a Nomenclature in the revised paper. Again, the Nomenclature is, in my opinion, required.

Author’s response:

Revised.

2.My previous comment 6 – From Figure 5 is clear that the Authors have observed 5 different cases. In the paper text related to Figure 6 is still missing an explanation – for which of five observed cases the grid verification is performed.

Author’s response:

Revised.

3.My previous comment 12 – A recent literature is not added. Only one reference is changed in comparison to the previous paper version. Used References are still dominantly older than 5-10 years. Therefore, a proper and recent literature addition is required.

Author’s response:

Revised.

Round 3

Reviewer 1 Report

I understand the scope of the manuscritp and require to discuss and investigate it in detail as a future work. 

Reviewer 3 Report

The Authors have preformed mentioned corrections/additions. The paper can be published in a presented form (after second revision).

This manuscript is a resubmission of an earlier submission. The following is a list of the peer review reports and author responses from that submission.